# A Computer Vision-Based System to Help Health Professionals to Apply Tests for Fall Risk Assessment

**DOI:** 10.3390/s24062015

**Published:** 2024-03-21

**Authors:** Jesús Damián Blasco-García, Gabriel García-López, Marta Jiménez-Muñoz, Juan Antonio López-Riquelme, Jorge Juan Feliu-Batlle, Nieves Pavón-Pulido, María-Trinidad Herrero

**Affiliations:** 1Clinical and Experimental Neuroscience (NiCE), Institute for Aging Research, Biomedical Institute for Bio-Health Research of Murcia (IMIB-Arrixaca), School of Medicine, University of Murcia, Campus Mare Nostrum, 30120 Murcia, Spain; jesusdamian.blascog@um.es (J.D.B.-G.); mtherrer@um.es (M.-T.H.); 2Automation, Electrical Engineering and Electronic Technology Department, Industrial Engineering Technical School, Technical University of Cartagena, 30202 Cartagena, Spain; gabriel.garcia@edu.upct.es (G.G.-L.); marta.jimenez@edu.upct.es (M.J.-M.); jantonio.lopez@upct.es (J.A.L.-R.); jorge.feliu@upct.es (J.J.F.-B.)

**Keywords:** fall risk, telemedicine, early diagnosis, RGB-D sensor, automation, gait, balance, tests, elderly, digital transformation of health systems

## Abstract

The increase in life expectancy, and the consequent growth of the elderly population, represents a major challenge to guarantee adequate health and social care. The proposed system aims to provide a tool that automates the evaluation of gait and balance, essential to prevent falls in older people. Through an RGB-D camera, it is possible to capture and digitally represent certain parameters that describe how users carry out certain human motions and poses. Such individual motions and poses are actually related to items included in many well-known gait and balance evaluation tests. According to that information, therapists, who would not need to be present during the execution of the exercises, evaluate the results of such tests and could issue a diagnosis by storing and analyzing the sequences provided by the developed system. The system was validated in a laboratory scenario, and subsequently a trial was carried out in a nursing home with six residents. Results demonstrate the usefulness of the proposed system and the ease of objectively evaluating the main items of clinical tests by using the parameters calculated from information acquired with the RGB-D sensor. In addition, it lays the future foundations for creating a Cloud-based platform for remote fall risk assessment and its integration with a mobile assistant robot, and for designing Artificial Intelligence models that can detect patterns and identify pathologies for enabling therapists to prevent falls in users under risk.

## 1. Introduction

The world is experiencing an increase in life expectancy. This brings with it the growth of the number of older people (60 years or more) who, according to the projections [1] of WHO (World Health Organization), will represent a sixth of the world population by the year 2030. Of course, this is a challenge from both the social and economic points of view, having to ensure that this population receives adequate health care tailored to their needs.

One of the problems that should be addressed are falls. Each year there are 37.3 million falls that require medical attention, and people over 60 are the ones who suffer them the most, becoming fatal [2] in many cases. In addition, the most frequent consequence of falls is usually hip fracture, which is associated with a decrease in functional capacity and a high risk of subsequent death, as indicated by different studies [3,4,5]. On the other hand, an investigation relates the fear of falling with a low HRQoL (Health Related Quality of Life) rate, both physically and mentally [6].

Also, the impact of falls on hospital pressure has been evaluated, and it has been found to affect women more than men [5,7,8]. With all this, falls represent a high economic cost, not only due to hospitalization, but also due to the subsequent dependence that it generates in the people who suffer them [9].

Faced with this situation, some countries have implemented prevention programs and their profitability has been increased [10,11]. In particular, it should be noted that in New Zealand, a home-based method has been found to be the most cost effective [12].

Once the problem of falls has been exposed and contextualized, it is important to review the clinical tools available to assess this risk. In the recent literature, the most used methods are the Tinetti test [13,14], the Berg scale [15,16], the Dynamic gait index [17,18,19], the test timed up-and-go (TUG) [20,21], the four-square test [22,23], the functional reach test [24,25], and the single leg stance test [26,27]. All of them have in common that they require the presence of a therapist responsible for evaluating and supervising the process [28].

The COVID-19 pandemic has shown the existence of problems in health attention when online platforms are needed for providing them [29,30]. Once COVID-19 risks and consequences have been dramatically reduced, health systems are mostly working again in a presential way. However, the adversities suffered by both clinicians and patients [31,32] during pandemics have revealed that the digital transformation of health systems is a pending subject, and a hybrid model capable of offering health solutions based on presential attention and the use of online tools could improve healthcare for all actors, both from a social and economic point of view. This is essential when patients are senior citizens with problems to be moved to hospitals or health centers from, for example, elderly institutions [33,34].

Therefore, there exists a need to automate diagnosis and treatments (even in the context of rehabilitation and fall risk monitoring, and this is what some investigations propose. The vast majority of them have chosen to integrate IMUs (Inertial Measurement Units), to obtain signals and associate them with the score of one of the previously described tests [35,36,37,38,39,40,41,42,43,44,45,46,47,48,49,50]. This solution has some drawbacks and limitations, since it is required that people wear the equipment on different parts of the body. On the other hand, it is usually associated with a single test (mainly the TUG), and does not provide information about other gait and balance parameters that other tests do take into account. Other authors use potentiometers [51], although they share the limitations of IMUs. Therapies with exoskeletons [52,53,54] have also been developed, albeit for rehabilitation and training purposes and digitally assisted through screens [55].

On the other hand, some investigations use Computer Vision techniques [56,57,58,59,60], while others have gone a step further and implement robots equipped with cameras and/or sensors. In the study described in [56], a person performs three types of exercises, and it is recorded by a camera located in a robot that follows a rectilinear trajectory, but the results are not correlated with any of the previously exposed tests, and many parameters of interest are not measured. In [61], a walker has been created for obtaining information during the TUG test, mainly through accelerometers. In [62], a system composed of both an inertial sensor and a robot with a camera has been developed, but is limited to measure the lower segment of the body.

Other interesting works are focused on validating the accuracy of human motion tracking by comparison with respect to stereophotogrammetric systems since, from the biomechanics literature, a stereophotogrammetric system represents a gold standard technique used for validating technologies like IMU or depth camera-based systems [63,64,65,66,67]. This is useful where the main purpose of the system is the accurate human motion tracking.

However, if the objective is to help therapists to apply tests such as Tinetti or TUG, where such therapists evaluate motion and poses in a coarse way, other cheaper and less invasive technologies could be suitable, despite losing a certain degree of accuracy, since the acquired data would be useful for early screening.

As the state-of-the-art reveals, it is very difficult to find a digital solution for helping therapists to continually monitor and evaluate the risk of falls in elderly people and, additionally, to estimate their progress in terms of increment or decrement of such risk of falls using ICTs (Information and Communication Technologies), since most of solutions found in the literature barely implement algorithms that take measurements about motion or pose in a usable and accessible way and present results in a comprehensible manner for health professionals.

Therefore, designing a digital platform capable of enabling such professionals to monitor and assess motion and gait difficulty, balance, and posture maintenance, even remotely, would be very useful for preventing certain events related to falls in elderly people. Furthermore, the sensors used for acquiring raw information about body pose and motion should be minimally invasive, avoiding forcing people to wear accessory devices, as far as possible, reducing the difficulty to carry out the tests and providing an objective way of assigning scores for each item of such tests.

Providing such a platform is the main goal of the work described in this paper. In particular, the system consists of a hardware device equipped with an RGB-D sensor that allows the patients’ body pose landmarks to be captured, from which a wide number of useful parameters are calculated, which could be stored both locally and in a Cloud-based platform. Such parameters are closely related to the items used by many well-known clinical tests of gait already described in the standard of care. Therefore, regarding previously published studies, this paper presents a comfortable solution for patients, since it is not necessary for them to wear any sensor; functional, since it can be used to assess most tests; and comprehensive, since the results are automatically generated and accessible by health specialists in the context of telemedicine.

The outline of the rest of the paper is as follows: Section 2 describes the materials and methods used and designed for developing the proposed system. Section 3 shows the tests carried out in both laboratory and real scenarios and the obtained results, and it describes the main advantages and limitations of the system in detail. Section 4 explains the limitations and measures for overcoming them and those open issues which are already being or are expected to be addressed in near future. Section 5 presents the conclusions.

## 2. Materials and Methods

This section describes the materials and methods used and developed for implementing the proposed system. The hardware and software architectures are detailed. Furthermore, as the MediaPipe Pose solution [68] has been used for landmarks’ detection, understanding such landmarks as relevant points in the image related to body joints, the process for calculating relevant parameters from such joints is explained. Such a set of parameters is useful for describing motion, gait, and balance, and they are related to the items found in many balance and gait tests found in the standard of care.

### 2.1. Hardware Architecture

Instead of using wearable devices based on IMUs, the proposed system uses a Computer Vision system, which could be easily placed at any location, in front of the elderly person whose motion and posture should be evaluated. This facilitates the installation of the system, and it is possible to use it in many scenarios with many patients if needed.

In fact, the authors of the paper are enrolled in a research project named JUNO+, funded by the Science and Innovation Ministry of Spain, during the years 2023 and 2024. One of the main objectives of such project is to deploy an assistant autonomous smart mobile robot capable of helping elderly people and their caregivers to do some simple tasks, such as attending a videocall, carrying out cognitive stimulation exercises, or monitoring the physical state of patients (including motion capabilities and balance), among others.

As the robot is already equipped with a touchscreen, an RGB-D sensor, and a Single Board Computer (SBC) for computation purposes, the hardware architecture of the proposed system in this paper specifically consists of such elements. Thus, the system developed for preventing falls will be easily integrated as an additional skill of the robot. Figure 1 shows how the proposed Computer Vision system would be installed in the robot.

Regardless of whether such Computer Vision system is included or not as a part of the robot, it works as follows: First, the information about a specific test is shown to the therapist for selection (see Figure 1a). Then, a specific item of the test could be chosen (each item involves the measurement of one or more parameters from the data obtained after body pose landmarks acquisition). The RGB-D sensor is used for recording a set of frames from which such landmarks will be calculated (see Figure 1b). Once this recording is finished, the set of calculated parameters related to the item are locally stored, and they could be eventually sent to the Cloud if needed (see Figure 1c). The set of computed parameters are presented as sequences.

Several advantages of using an RGB-D sensor have been taken into account compared to the usage of other vision systems. First of all, the economic cost of this kind of sensor is suitable, considering the capacity to estimate distances and to obtain accurate 3D maps of the environment and objects located to distances smaller than 5 m. Monocular cameras are cheaper, but they do not provide 3D mapping. On the other hand, stereo vision systems do not provide much better accuracy in 3D reconstruction, but they are more expensive. As the system will be used in indoor environments, the advantage of being used outdoors, which stereo vision systems present, is not relevant in this case.

### 2.2. Software Architecture

The software architecture has been designed as a set of software modules (see Figure 2), which work in ROS (Robotic Operating System) [69], since, as mentioned before, the system should work both as an independent hardware device or as a part of the robot JUNO in the JUNO+ project. For acquiring frames from the RGB-D sensor (whose model is an Orbbec Astra Pro), the ROS package provided by the manufacturer is used. When the appropriate ROS node from this package (*camera node*) is executed, several topics are exposed, and the information about each RGB frame and its corresponding Depth image are published in the topics */camera/imageRGB* and */camera/imageDepth*. Both images are registered and synchronized, so it is possible to easily assign a distance to each pixel. Moreover, information about the calibration of both cameras (one for acquiring color images and the other one used for depth calculation) is also available through the ROS topics */camera/rgbInfo* and */camera/irInfo*.

A client node (named *client_pose*) is subscribed to the mentioned topics for obtaining the RGB image and its corresponding depth information each time. The RGB image is then sent as a parameter of a request made to the service node (*/srv_mediapipe*). Once the request is received, the service node returns a response coded as a “pose” type message that contains a bidimensional array of points (in the coordinate system of the RGB image), corresponding to the pixels where a person’s joints are located according to the landmarks shown in Figure 3.

### 2.3. Using the MediaPipe Pose Solution

The service node uses the MediaPipe Pose solution, which is based on a Deep Learning algorithm capable of obtaining 33 joints of the human body (see Figure 3) using an RGB image as input. Each obtained point representing a joint is coded as a pair (column and row) normalized in the interval [0, 1]. For projecting each pair on the original RGB image, the column and row should be multiplied by number of columns and number of rows of such image, respectively.

Once the position of the joints within the image is obtained, the client node, using the depth channel of the camera and knowing the position of the pixel, finds the real depth associated with those joints according to:(1)$$z=depth\left(i,j\right)\;$$
where *depth (i*, *j)* is the depth value at the row *i* and column *j*. Note that the *(i*, *j)* coordinates correspond to the same pixel in both images RGB and Depth.

However, the points are not referenced to any three-dimensional coordinate system. To do this, the RGB-D camera has been calibrated to obtain its intrinsic parameters and place the points in a system that has the camera as its origin. The calibration matrix M is a 3 × 3 matrix:(2)$$M=\left(\begin{array}{ccc} f_{x} & 0 & c_{x}\\ 0 & f_{y} & c_{y}\\ 0 & 0 & 1 \end{array}\right),$$
where *f_x_*, *f_y_*, and *c_x_*, *c_y_* are the focal length and the optical centers, respectively. The 3D coordinates are computed using the following formulas:(3)$$x=\frac{\left(j-c_{x}\right)\times z}{f_{x}}\;$$
(4)$$y=\frac{\left(i-c_{y}\right)\times z}{f_{y}}$$
(5)z=depth(i,j)
where the pair *(i*, *j)* represents the coordinates of a pixel.

Note that, although the MediaPipe Pose solution is capable of inferring depth from the 2D image, this depth is referenced to the center of the body, and does not allow the reconstruction of the 3D space according to the reference system of the RGB-D sensor. As it is particularly important the computation of the speed and the trajectory of the subject for estimating the score of some items related to gait in most of standard of care tests, values obtained from the MediaPipe Pose solution are transformed to the RGB-D sensor reference frame by applying the corresponding transformation by using Equations (3)–(5).

It is important to highlight that it is assumed that the patient should be properly located at a specific initial position from which each exercise is started. Therapists or caregivers are responsible to guide the patient, to avoid occlusions and to ensure the appropriate visibility, for this early prototype. Therefore, the algorithm considers only one detected user as the target one.

### 2.4. Parameters’ Computation

From the three-dimensional position of the joints already calculated with respect to the camera, it is possible to make multiple measurements that will be used to evaluate the different items of the gait and balance tests. Such measurements are the following:Speed: To measure speed, the midpoint of the hip is taken and the displacement it experiences is calculated in 1-s time intervals.Trajectory. To measure the trajectory, the midpoint of the hip is taken and projected onto the ground.Trunk swing (see Figure 4a): It is measured through the angle of inclination of the shoulders.Separation of the arms (see Figure 4b): The separation of the arms with respect to the trunk is calculated from two angles, one for each side. Using the points of the elbow, shoulder, and hip.Arm support (see Figure 4c): To measure whether the user uses their arms to get up from a chair, the angle formed by the shoulder, elbow, and wrist is calculated.Separation of the heels (see Figure 4d): The horizontal distance between the ankle points is calculated.Stride (see Figure 4e): To measure the amplitude of the steps, the depth difference between the ankle joints is calculated.Hand separation (see Figure 4f): The movement of the hands while walking is measured by calculating the distance between the wrists.Trunk tilt (see Figure 4g): To measure trunk tilt, the angle formed by the shoulder, hip, and knee is measured. Both on the right and left side.

The formula used to calculate the angles formed by three joints is as follows:(6)$$\alpha =\cos^{-1}\frac{\left|\vec{AB}\cdot \vec{AC}\right|}{\left|\vec{AB}\right|\cdot \left|\vec{AC}\right|}\;,$$
where A, B, and C are the three-dimensional points corresponding to the joints. Figure 5 shows an example for calculating the separation of the arms.

### 2.5. Relation between Parameters and Items of Tests Used for Gait and Balance Evaluation

Table 1 shows the minimum number of parameters applicable to each of the items evaluated in many tests found in the standard of care.

For example, the Tinetti test consists of 16 items where balance (from 1 to 10) and gait (from 11 to 16) are evaluated. Item 1 allows therapists to evaluate balance when the person is seated in a chair. In this case, according to the tilt of the trunk, it is possible to estimate if he/she is correctly seated and if he/she is properly keeping the balance. Consequently, only the parameter trunk tilt would be measured for automatically assessing such item. However, item 9 allows therapists to evaluate how a person sits in a chair, that is, the process from standing to be seated. In this case, the trunk swing and tilt, together with the arm support are relevant parameters for automatically assessing the behavior of the patient. For instance, a person who needs to use the arms to get seated probably should get a worse score than who does not need them.

## 3. Results and Discussion

A wide number of different tests have been carried out in a controlled laboratory environment for validating the designed algorithms. In this case, a person has carried out different motions (some of them are forced), to simulate the behavior of people with different ability for motion and balance. In them, each of the parameters has been individually analyzed. Once such tests have been made, the system has been validated in a real scenario with real patients. Several sequences that provide information about the time evolution of each parameter are shown and analyzed in the following subsections. How to interpret such sequences is also explained in detail.

### 3.1. Controlled Laboratory Tests

The trunk swing, measured from the angle of the shoulders, is depicted in Figure 6. This is the parameter most associated with balance, and is easily identifiable; the loss of stability immediately causes the trunk to sway and, in the worst case, a fall. In the first capture, a straight position offers values lower than 5°, while in the second capture there is a marked trunk swing with angles greater than 35° (in this case, the pose has been forced for simulating a strong swing of the trunk).

Note that the Y-axis represents angle degrees, and the X-axis represents samples across time.

Separating the feet increases stability, consequently, it is a resource used by those people who suffer from balance problems, since they separate their feet further to find a safe position. In Figure 7, the distance between heels has been measured; in the first capture, the feet are separated at half a meter, while in the second, the feet are together ten centimeters apart. In this case, the Y-axis represents meters, and the X-axis represents samples across time.

The use of the arms is an innate resource to improve balance. Figure 8 shows the evolution of the angle of the arms, both right and left. In the first capture, the right arm remains up, while the left is down. In the second, both arms are raised, forming an angle of almost 90° with respect to the trunk. In this case, the Y-axis represents angle in degrees, and the X-axis represents samples across time.

The user trajectory has been represented in Figure 9. The horizontal axis of the graph represents the depth of the subject relative to the camera, and the vertical axis represents its horizontal position. Path 1 represents a rectilinear path and path 2 represents a curved deviation trajectory. The oscillations that can be seen in trajectory 1 correspond to the natural movement of the body when walking, which moves with small horizontal movements that are captured by the camera.

Short steps while walking provide stability, while long steps can cause greater imbalances, so people with mobility problems opt for a shorter stride. Figure 10 shows the size of the step during walking, calculated from the difference in depths of the points of the feet. The size of the step is measured at the ends of the signal, which is when both feet are already flat on the ground. In the first sequence, the steps are about 20 cm, while in the second, the steps reach 40 cm. In this case, the Y-axis represents meters, and the X-axis represents samples across time.

Getting up from a chair and sitting down is not a simple task when there are mobility problems. The number of attempts and successes, as well as whether the person holds on to something to achieve it, is used by various tests to measure the patient’s fragility. The evolution of the angle of the trunk has been represented in Figure 11. This parameter is useful to analyze when the subject stands up or sits in the chair. There are two signs, corresponding to each side of the trunk, and a third signal representing the mean. In the first capture, the subject is seated, and his angles are around 120°. In the second one, the subject has stood up and its trunk forms an angle greater than 160°.

In the previous trunk inclination test, it is also possible to measure whether the user uses his arms to stand up or not. Both situations have been represented in Figure 12. In the upper capture, the patient keeps his arms extended during the transition to standing. In the lower capture, the subject uses his arms to stand up and is thus reflected in the angle of the arms, which are no longer extended and are bent at angles less than 100°.

### 3.2. Testing in a Real Environment

Once the laboratory tests were completed, an experiment was carried out in a nursing home (see Figure 13), in which six patients participated (three women and three men).

Four exercises were performed, based on different balance and gait scales. In Figure 14, the evaluated items are shown together with the parameters evaluated during their execution.

Note that the patients were selected by the institution’s staff, considering people with a similar ability to do the chosen exercises, without relevant pathologies related to balance, in order to better validate the capacity of the system for identifying patterns in the sequences which are probably indicative of problems with balance according to items and parameters described in Figure 14.

Next, the most significant results of the different tests will be pointed out and comparisons will be made between patients.

Figure 15 shows the graphic representation of sequences that represent sitting and standing tests for patient 2. Figure 16 shows the obtained results for patient 4, doing the same exercise. The different positions (standing or sitting) and the transitions between both are easily identifiable. Furthermore, the angle of the arms allows to know that patient 4 was able to keep his arms extended during the test, while patient 2 needed to support them. Thus, the evolution of the angle of the arms of patient 2 evolves at the same time as the movements of the trunk, extending when he is standing and bending when he remains seated. While in patient 4 no variations were observed.

For the balance test (see Figure 17), patients had to remain still in the same position for a few seconds. It should be noted that all patients responded well, but small differences can be seen. In the case of trunk sway, patient 3 barely moved, while patient 1 experienced slight trunk movements. Regarding the separation of the heels, patient 4 managed to maintain a distance 10 cm shorter than patient 2 without this affecting their balance.

In the same position as the previous test, patients were required to close their eyes (see Figure 18). Once again, all responded in a similar way; subject 1 continued to be the one with the least trunk sway and patient 3 the most. Regarding the distance of the heels, patient 2 kept his feet the furthest apart and patient 6 was the one who separated them the least. None of the participants helped themselves with their arms.

During the walk, in which the patients covered approximately 4 m, the trajectory followed the separation of the arms, the swing of the trunk, the stride, and the movement of the hands were calculated. Regarding the trajectory (see Figure 19), it can be seen how patient 1 was able to stay straight, while patient 5 deviated at the end of the path.

Regarding the swing of the trunk and the separation of the arms, there were no major differences or notable events. However, it is interesting to note how some patients move their hands alternately to their feet in the stride, such as patient 2. While others hardly used their hands, this was the case of patient 4 (see Figure 20). Abnormal movements, such as only moving one of the arms, could be symptomatic of a neurodegenerative disease.

The results obtained both in the laboratory and later, in the real setting, confirm the value of the tool to provide useful and objective information to therapists about the different gait and balance tests. This solution enables a posteriori evaluation, inasmuch as the tests could be carried out in the company of an assistant and the results could be analyzed by the specialist later. Furthermore, the classic performance of the tests does not provide precise or quantified information. While this solution allows that when a longitudinal study of the patient is carried out, the sequences could be compared, and progress measured. On the other hand, it has the advantage of not needing to equip the patient with any sensor, they only have to perform the exercises in front of the camera.

## 4. Limitations and Future Enhancement

Since the work presented in this paper is an early prototype, several limitations could be highlighted, together with the specific measures to carry out for overcoming such limitations:Currently, the system is ready to be mainly used in elderly institutions. The deployment of the system needs the involvement of some institution’s workers. Preventing the overload of such members’ work tasks is a goal for getting a better acceptance of the system. Therefore, the prototype will be used as a part of the rehabilitation programs existing in the institution. In the near future, when several robots are deployed as a result of the research project JUNO+, an improvement will be developed to allow assistant robots to autonomously guide the procedure for executing the tests. In this case, the therapist will only have to design the set of exercises that are included in each evaluated item (in a specific test) and to schedule the activity as a part of the rest of help tasks that the robot needs to carry out.The prototype has been designed after analyzing the needs and solutions found in the literature. However, it would be interesting to empirically study how the need for the digital transformation of health systems is perceived by clinicians (mainly in the context of elderly people monitoring and rehabilitation). In the near future, a usability study, which allows the comparison between applying tests in a traditional way and by using the proposed system, will be carried out. This usability test will be used to improve the prototype, adapting the user interface to the needs of therapists and caregivers (both formal and informal). Thus, the system could be used outside of the elderly institutions, for example with patients who live at home and are cared for by informal caregivers (for instance relatives).Table 1 shows the mapping between items and parameters calculated from raw data acquired the RGB-D sensor. Such mapping has been designed by the researchers, but it is expected to be empirically validated by clinicians and adapted, if needed, in a future work.

On the other hand, beyond the measures that will be considered to overcome the detected limitations, authors are currently working in improving the system. In particular, they are extending it to a Cloud-based platform, where all the data about patients monitored by a therapist would be properly saved. The therapist will also be able to design different tests consisting of different items. Such tests will be applied as many times as needed with the same patient, allowing the therapist to evaluate the progression in gait and balance for such patient.

As aforementioned, the proposed system is designed for a coarse estimation of body pose for helping therapists to assess fall risk detection through tests such as Tinetti or TUG. MediaPipe offers a high rate of success in suitable identification of the main joints of the human body [70,71,72]. In fact, the MediaPipe solution has been tested by its developers by using a Pose Validation dataset, yielding a PDJ (Average Percentage of Detected Joints) of 97.5%. PDJ is a strong indicator of precise matches between predicted keypoints and ground truth keypoints.

However, this model used for human motion tracking is not validated with respect to a stereophotogrammetric system yet, and the biomechanics literature describes stereophotogrammetric systems as the gold standard technique used for validating other technologies like IMU-based systems or depth camera systems. Therefore, this kind of validation will be carried out as future work.

As mentioned before, a trial with a limited number of patients has been carried out, but, in the future, it should be extended to a larger population and establish which of the parameters are most significant to identify fall risk patterns, in collaboration with health professionals. Once analyzed using the best parameters according to the knowledge of a wide number of health professionals that use the future platform, it is expected to develop an Artificial Intelligence-based agent capable of providing scores for each item automatically, helping therapists to carry out faster screening tasks useful for preventing falls more efficiently.

Finally, as the system has been designed to be fully compatible with the robot JUNO, such a robot could be also used for non-intrusive recording the movements and poses of residents in nurse homes, who should be continuously monitored, constantly extracting and analyzing gait and balance parameters. If any pathology is detected, it could send a warning to healthcare personnel so that they could take the appropriate measures. In this way, progress would be made in the necessary prevention and automation of early diagnosis. 

## 5. Conclusions

The proposed solution implements a digital tool capable of helping health professionals to carry out and evaluate balance and gait tests, which are well known in the standard of care. Therapists can easily analyze the resulting sequences by using a graphic representation of them, to establish a more precise diagnosis and to monitor the temporal evolution of patients. On the other hand, the solution also allows raw data (body pose landmarks) and aggregated information (parameters) to be recorded. This helps the simultaneous presence of the therapist and the patient to be avoided if needed, since patients could carry out the exercises accompanied by an assistant, and the obtained results could be examined later by the therapist.

Unlike other studies that focus on a single test, a generic system applicable to any test is provided here. Furthermore, the patient does not have to wear any sensor, which is a great convenience. On the other hand, the obtained information could be adequately stored in the Cloud and presented to health specialists, guaranteeing security and privacy standards.

## Figures and Tables

**Figure 1 sensors-24-02015-f001:**
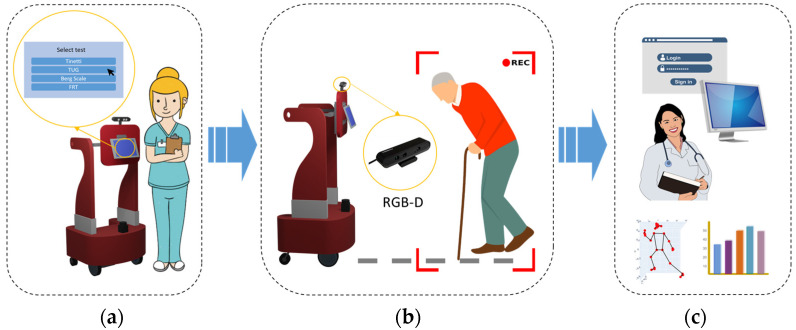
Description of the proposed system installed as a part of the JUNO robot. (**a**) Interface in a touchscreen for selecting a test, found in the standard of care, for evaluating balance and gait. (**b**) Recording the elderly people’s motion through an RGB-D sensor for body pose landmarks calculation. (**c**) Storing parameters as sequences for its evaluation locally or in the Cloud.

**Figure 2 sensors-24-02015-f002:**
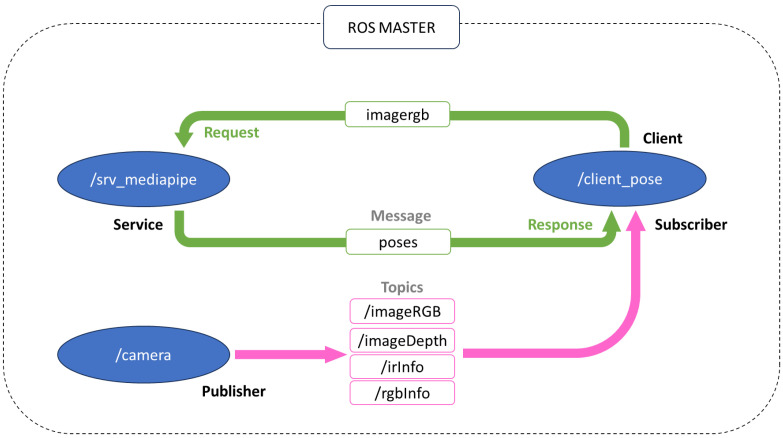
Outline of the software architecture as a set of components that share information through ROS.

**Figure 3 sensors-24-02015-f003:**
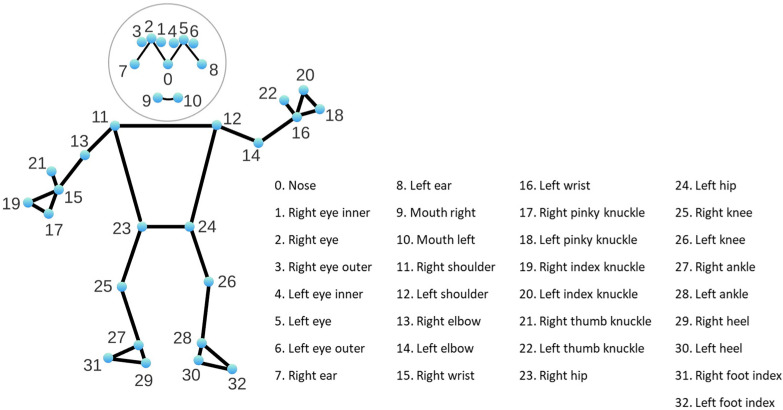
Body poses landmarks obtained when the Pose solution from MediaPipe is used. This graphical representation has been obtained from the MediaPipe website [68].

**Figure 4 sensors-24-02015-f004:**
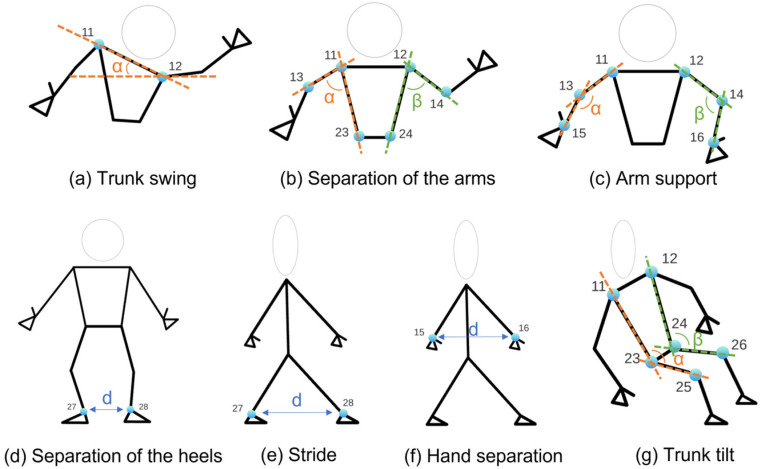
Set of parameters calculated from the set of joints (3D points in the RGB-D sensor reference system), obtained after applying the MediaPipe Pose solution.

**Figure 5 sensors-24-02015-f005:**
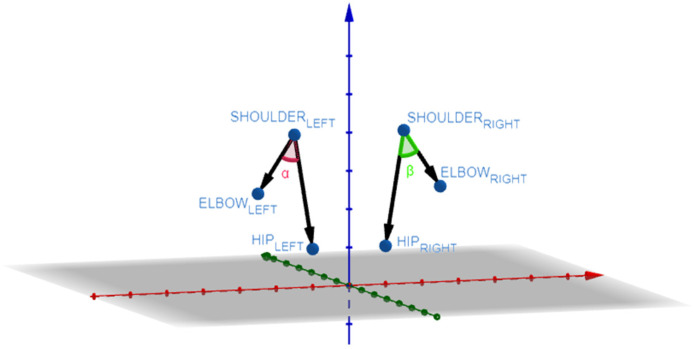
3D points used for calculating the angle between the arms and the trunk.

**Figure 6 sensors-24-02015-f006:**
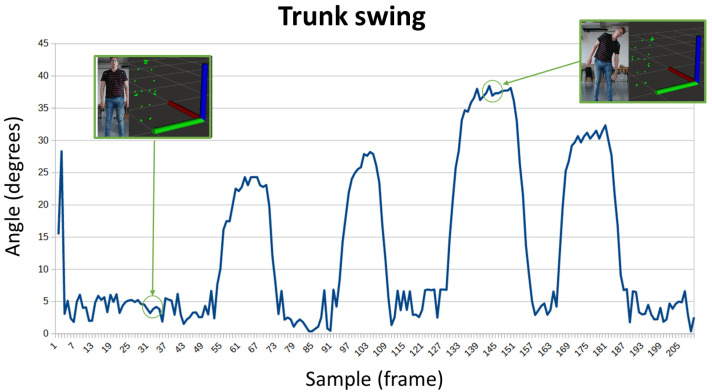
Evolution of the trunk swing while a person is balancing the trunk.

**Figure 7 sensors-24-02015-f007:**
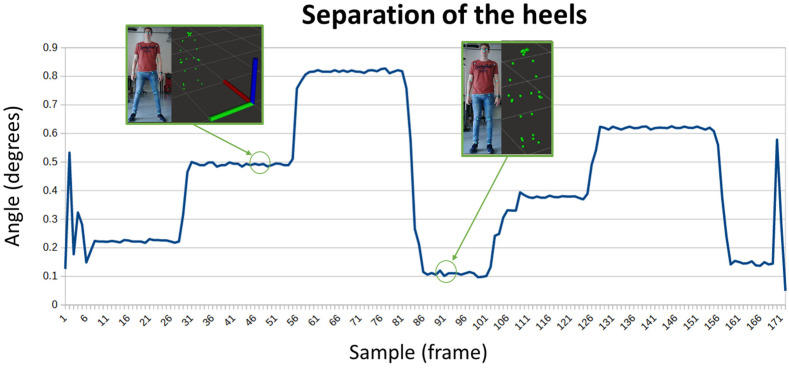
Evolution of the separation of the heels while a person changes the position of the legs.

**Figure 8 sensors-24-02015-f008:**
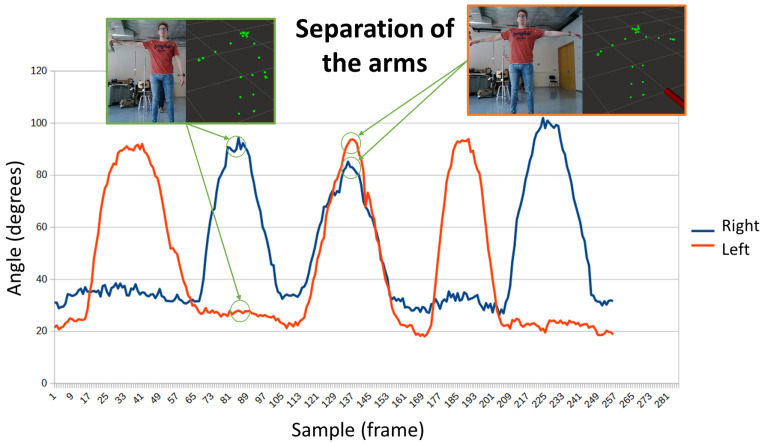
Evolution of the separation of arms. The person first raises the right arm and later he raises the left arm.

**Figure 9 sensors-24-02015-f009:**
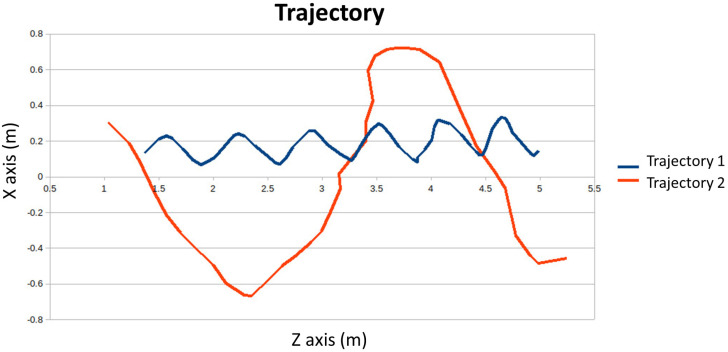
Trajectories when a person is walking. This parameter is related to items that allow gait to be evaluated.

**Figure 10 sensors-24-02015-f010:**
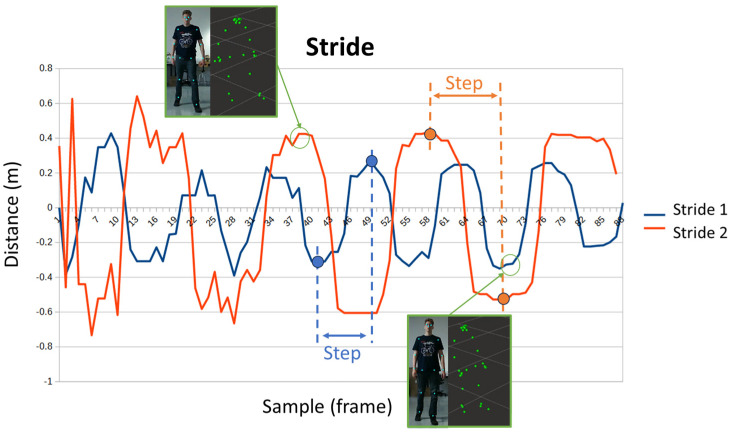
Evolution of the stride while a person is walking.

**Figure 11 sensors-24-02015-f011:**
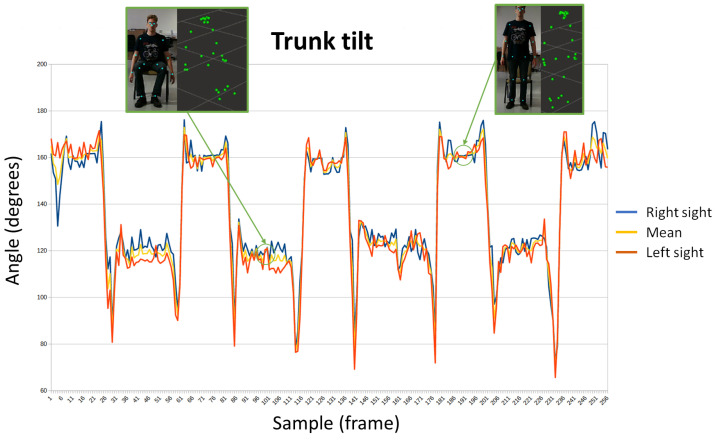
Evolution of the trunk tilt while a person stands ups or sits in a chair.

**Figure 12 sensors-24-02015-f012:**
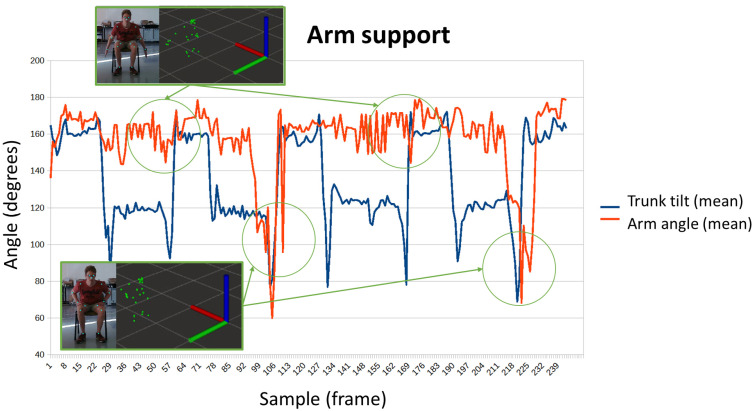
Evolution of the position of the arms (represented by the angle) and the trunk tilt while a person is sitting in a chair. This allows therapist to know if the arm support is used, by analyzing the combination of the two time-series.

**Figure 13 sensors-24-02015-f013:**
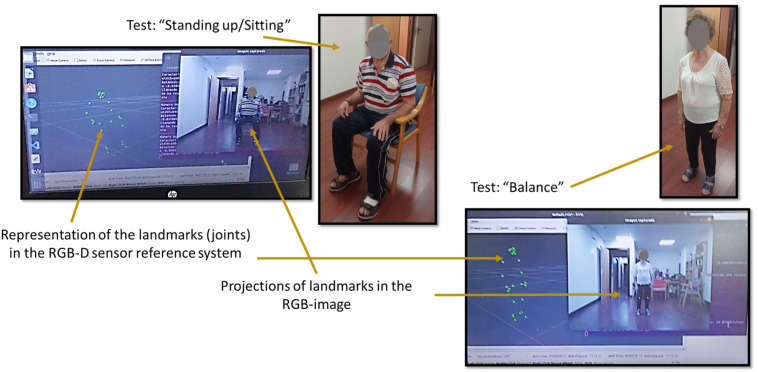
Two patients who collaborated in the tests. Larger pictures have been taken by the researcher that carried out the tests. The other pictures represent the point of view of the RGB-D sensor.

**Figure 14 sensors-24-02015-f014:**
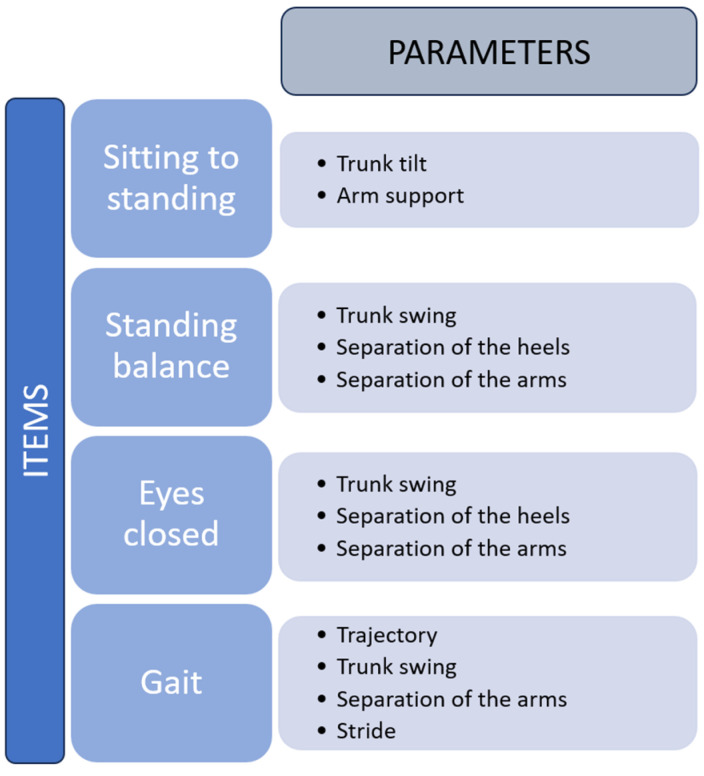
Items and parameters evaluated in a real environment.

**Figure 15 sensors-24-02015-f015:**
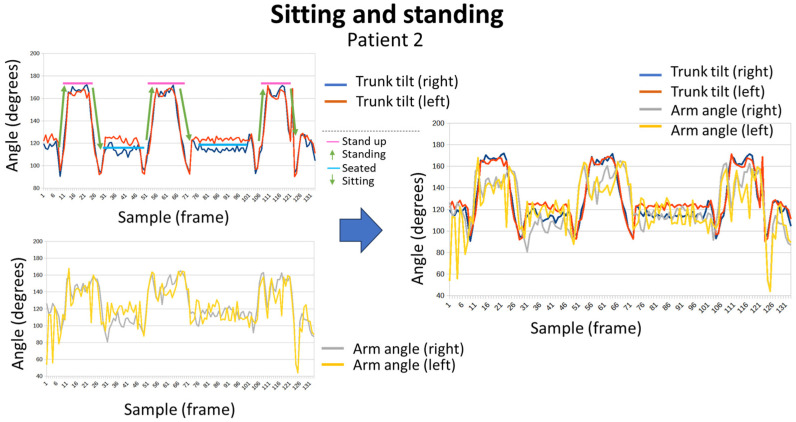
Sitting and standing exercise for patient 2.

**Figure 16 sensors-24-02015-f016:**
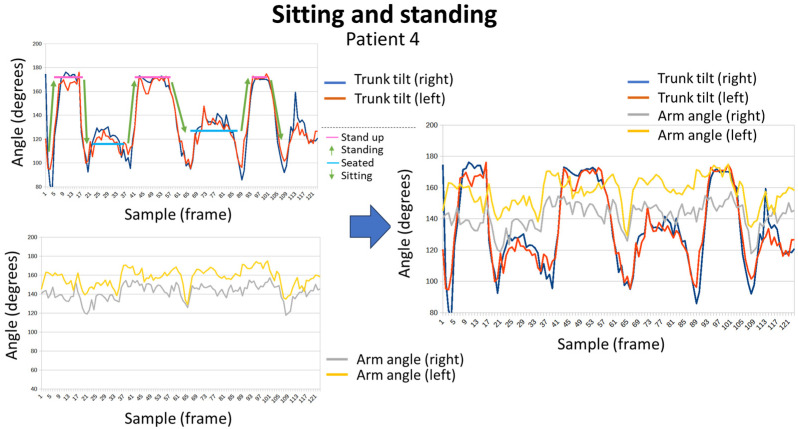
Sitting and standing exercise for patient 4.

**Figure 17 sensors-24-02015-f017:**
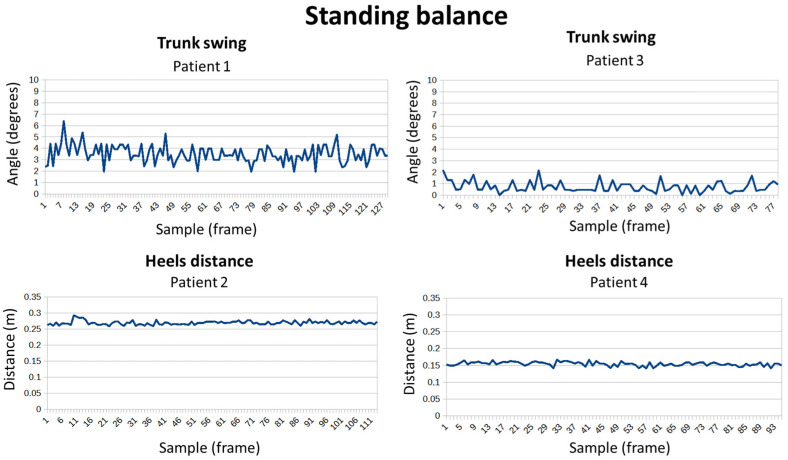
Standing balance exercise for four patients.

**Figure 18 sensors-24-02015-f018:**
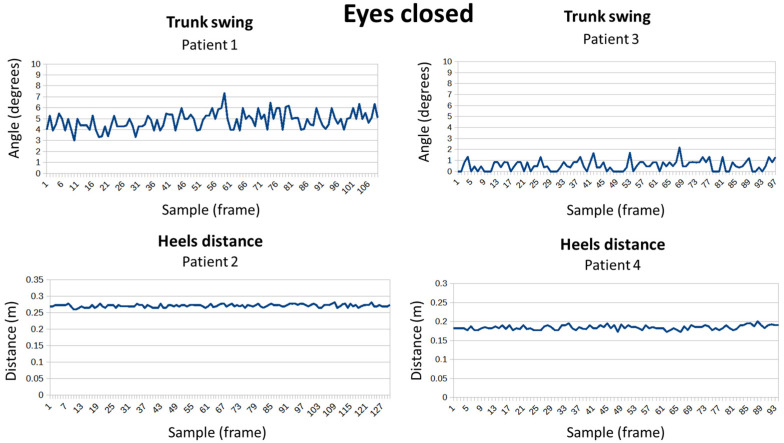
Measuring balance with eyes closed for four patients.

**Figure 19 sensors-24-02015-f019:**
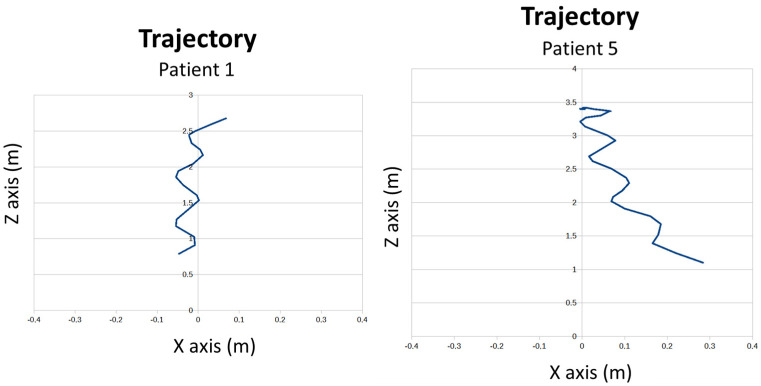
Trajectories for patient 1 and patient 5 during gait.

**Figure 20 sensors-24-02015-f020:**
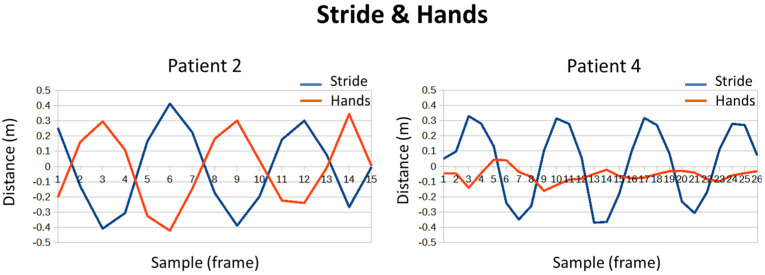
Stride and hand’s movement during gait.

**Table 1 sensors-24-02015-t001:** Parameters applicable in the different tests and items.

		Parameters								
		S	T	TS	SA	SH	S	TT	HS	AS
TINETTI ITEMS	1							×		
	2							×		×
	3							×		
	4			×		×				
	5			×		×				
	6			×		×				
	7			×		×				
	8		×	×		×	×			
	9			×				×		×
	10		×							
	11						×			
	12						×			
	13						×			
	14		×							
	15		×	×	×					
	16					×				
BERG ITEMS	1							×		×
	2			×						
	3							×		×
	4							×		×
	5							×		×
	6			×						
	7			×		×				
	8				×			×		
	9							×		
	10			×						
	11	×	×							
	12			×			×			
	13					×	×			
	14			×			×			
DGI ITEMS	1	×	×	×						
	2	×	×	×						
	3	×	×	×						
	4	×	×	×						
	5	×	×	×			×			
	6	×	×	×						
	7	×	×	×			×			
	8						×			×
TUG		×	×							
FRT								×	×	
FSST		×	×			×				
SLS						×				

S—Speed, T—Trajectory, TS—Trunk swing, SA—Separation of the arms, SH—Separation of the heels, S—Stride, TT—Trunk tilt, HS—Hand separation, AS—Arm support.

## Data Availability

Data are unavailable due to privacy restrictions.

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
