# Peer review of "A Computer Vision-Based System to Help Health Professionals to Apply Tests for Fall Risk Assessment"

_sensors, 2024, doi:10.3390/s24062015_

Round 1

Reviewer 1 Report

Comments and Suggestions for Authors

They propose a tool to automate the evaluation of gait and balance in elderly people and thus avoid or prevent possible falls. They do this by relying on a camera that captures movements that will then be analyzed by a therapist.

The proposed system has been validated both in the laboratory setting and with 6 elderly people with good results.

It would only be appreciated if they could indicate the possible ethical issues between the data collection provided by this camera, the data uploaded and shared in the cloud, and the privacy of individuals. 
Additional comments

  1. 1. The main question addressed by the research is how to automate the evaluation of gait and balance, essential for preventing falls in older people, using an RGB-D camera and digital representation of human motion parameters.

  2. 2. The paper presents an original approach to remotely evaluate gait and balance, which is particularly relevant given the challenges posed by the increasing elderly population. It addresses the gap in the field of providing a tool for automated evaluation that does not require therapists to be present during exercises.
    3. The label size for Fig. 1 is too small and the first column of Table 1 needs to be rearranged.

  3. 4. Compared with other published material, this study adds a practical demonstration of a system that can objectively evaluate key aspects of clinical tests related to gait and balance. It also suggests future applications, such as a Cloud-based platform for remote fall risk assessment and integration with mobile assistant robots, which could significantly enhance the ability to prevent falls in at-risk individuals.

Reviewer 2 Report

Comments and Suggestions for Authors

The study presents a computer vision-based approach for the joint angles of upper and possible lower segments. The development of the sensor and the technology is clear and also presented coherently in the methods. However, the study presents two main issues that are major points that need to be addressed in my opinion.

1)As first, the computation of the angles obtained by virtual markers recovered by the cameras does not follow the standard biomechanical convention. Instead, this is very important. Indeed the use of the standard kinematics conventions proposed by Wu and colleagues in 2002 and 2005 are essential to follow kinmatic analysis.

2) The authors did not use any stereophotogrammetric system to validate the actual measurement. Thus without a validation of the data with respect to the gold standard, it is difficult to support the conclusion.

ADDITIONAL COMMENTS  

1) Although the manuscript presents the development of a sensor and computational model for the development of gait analysis techniques, the model was not validated with respect to a stereophotogrammetric system. In particular, we know from biomechanics literature that a stereophotogrammetric system represents a gold standard technique used for validating other technologies like IMU-based systems or depth calera systems.

For this reason, I suggest the Authors to add a sterepammetic validation as also done for other computer vision-based systems of human motion tracking. An interesting and recent paper Authors are encouraged to review is :  

- Balta, Diletta, et al. "A model-based markerless protocol for clinical gait analysis based on a single RGB-depth camera: concurrent validation on patients with cerebral palsy." IEEE Access (2023).  

In the aforementioned paper, the role of concurrent validation with a gold standard method is proposed. Authors should thus present concurrent validation results and not only talk about validation over pathological subjects. The role of validation with respect to gold standard technologies indeed represents an important step in biomechanics and gait analysis. This is also the reason why the computational models of segment angles should be compared with models applied in gold standard protocols.

It is well known for people who work in human kinetics analysis that certain protocols have to be adopted for the analysis. ISB (International Society of Biomechanics)  showed two important consensus manuscripts for defining the joint angles that also have a functional meaning and they are adopted in the clinical field. Thus Authors should use standards in biomechanics as defined in the following:  

-Wu, Ge, et al. "ISB recommendation on definitions of the joint coordinate system of various joints for the reporting of human joint motion—part I: ankle, hip, and spine." Journal of Biomechanics 35.4 (2002): 543-548.  

-Wu, Ge, et al. "ISB recommendation on definitions of joint coordinate systems of various joints for the reporting of human joint motion—Part II: shoulder, elbow, wrist, and hand." Journal of Biomechanics 38.5 (2005): 981-992.  

So please when Authors refer to trunk angle, for instance, They should refer to the proper angle according to the opportune functional convention.  

2) Authors should use concurrent validity plots as the ones typically employed in concurrent validation approaches as the bland altman plots of the errors between the developed sensor and the gold standard technology. This is a very useful tool for supporting the Discussion that Authors advanced regarding the transferability of the instrumentation in the real scenario.  

3) Can the authors further elaborate on the statistical assessment they performed for assessing the validity of their kinematic models?  

4) In the introduction Authors should enlarge the perspective related to having good and reduced sensor setup for assessing human motion. This point has implications in many technological domains. For this reason, I suggest the Authors to review the following papers:  

-Mobarak, Rami, et al. "A Minimal and Multi-Source Recording Setup for Ankle Joint Kinematics Estimation During Walking using only Proximal Information from Lower Limb." IEEE Transactions on Neural Systems and Rehabilitation Engineering (2024).  

-Duraffourg, Clément, et al. "Real-time estimation of the pose of a lower limb prosthesis from a single shank mounted IMU." Sensors 19.13 (2019): 2865.  

END OF COMMENTS

Round 2

Reviewer 2 Report

Comments and Suggestions for Authors

I appreciate the Authros responses and their modifcation in the paper to stress that the validation with respect to a gold standard technology has to be done and this represent a limitation of the study. Although I strogly suggest the Authros to do a validation with respect to standard biomechanics protocol the preliminary assessment can be strenghten by consensus from clinicans who are employing the methodology. Thus I suggest the Authors to add in the discussion this aspect adding a sentence. In any case this is leave to the Authros and I think that the paper can be accepted.